# Somatic Reprogramming—Above and Beyond Pluripotency

**DOI:** 10.3390/cells10112888

**Published:** 2021-10-26

**Authors:** Yaa-Jyuhn James Meir, Guigang Li

**Affiliations:** 1Graduate Institute of Biomedical Sciences, College of Medicine, Chang Gung University, Taoyuan 333, Taiwan; 2Department of Biomedical Sciences, College of Medicine, Chang Gung University, Taoyuan 333, Taiwan; 3Limbal Stem Cell Laboratory, Department of Ophthalmology, Chang Gung Memorial Hospital, Linkou 333, Taiwan; 4Department of Ophthalmology, Tongji Hospital, Tongji Medical College, Huazhong University of Science and Technology, Wuhan 430030, China

**Keywords:** induced pluripotent stem cell (iPSC), somatic reprogramming, Col1a1 4F2A Oct4-GFP reprogrammable mouse, stochastic and deterministic model, expanded potential stem cell (EPSC), expanded potential stem cell medium (EPSCM)

## Abstract

Pluripotent stem cells, having long been considered the fountain of youth, have caught the attention of many researchers from diverse backgrounds due to their capacity for unlimited self-renewal and potential to differentiate into all cell types. Over the past 15 years, the advanced development of induced pluripotent stem cells (iPSCs) has displayed an unparalleled potential for regenerative medicine, cell-based therapies, modeling human diseases in culture, and drug discovery. The transcription factor quartet (Oct4, Sox2, Klf4, and c-Myc) reprograms highly differentiated somatic cells back to a pluripotent state recapitulated embryonic stem cells (ESCs) in different aspects, including gene expression profile, epigenetic signature, and functional pluripotency. With the prior fruitful studies in SCNT and cell fusion experiments, iPSC finds its place and implicates that the differentiated somatic epigenome retains plasticity for re-gaining the pluripotency and further stretchability to reach a totipotency-like state. These achievements have revolutionized the concept and created a new avenue in biomedical sciences for clinical applications. With the advent of 15 years’ progress-making after iPSC discovery, this review is focused on how the current concept is established by revisiting those essential landmark studies and summarizing its current biomedical applications status to facilitate the new era entry of regenerative therapy.

## 1. Introduction

The discovery of induced pluripotent stem cells, a monumental breakthrough, rewrote the conceptual foundation in biology. It endowed an unprecedented opportunity to unravel the process of cell identity establishment and potentiate cell regenerative therapy in clinical settings. Concomitantly with the union of gametes, a fertilized egg gradually specifies lineages at the expense of differentiation potency during developmental processes. By contrast, a highly differentiated cell can re-acquire its differentiation potency during the cellular reprogramming process. As maintaining a differentiated state is imperative to engage the proper function of the cell, it needs mechanisms to “lock on” the status of differentiation to warrant its cell identity. Such mechanisms of the differentiated state assurance are presumably an essential barrier for cellular reprogramming [1]. As a cell’s identity comes from its unique epigenome configuration, erasing and writing the epigenome for resetting its landscape leads to changing the cell fate. One of the main ways to reconfigure the epigenetic landscape is through DNA replication during cell division. Thus, cycle re-entry is an essential step of somatic reprogramming of the post-mitotic cell. In contrast, nuclear reprogramming through SCNT (somatic cell nuclear transfer) does not require cell cycle progression. Accordingly, the reprogramming process can be achieved in different routes through which a coordinated multifactorial and multistep process for sequentially re-gains the state of pluripotency, although it may not be necessary to reverse the same path as that taken by the differentiation process [2,3,4,5].

A considerable amount of effort has been put into investing these reprogramming mechanisms in the past half-century. Figure 1 shows the timeline of significant scientific advances in the history of nuclear and somatic reprogramming research. As a result, we have gained fruitful insights that further enlighten and set new stages for stem biology, permitting researchers to dive deeper into the dynamic translation process of genome plasticity and its corresponding cell state. In the present review, we revisited these essential studies established in the last 15 years and mainly focused on somatic reprogramming (an excellent review on SCNT can be found elsewhere by Matoba et al.). Here, we introduce the conceptualization of somatic reprogramming from the aspects of cell potency acquisition, route choice, and network-relaying in order to unfold its potential mechanisms. Furthermore, revisiting progress-making in the directed modifications of genome configuration, either in the direction of differentiation or de-differentiation, by acting through the nature of a cell’s genome plasticity, will help tackle emergent issues in future biology and their potential applications in clinical settings.

## 2. Genome Plasticity Endows Cell Fate Change

A multicellular organism’s development is a continuous process in diversifying cell functions from a single cell (i.e., zygote). The resultant differentiated cells are formed through a progressive procedure by restricting their developmental potential and, in turn, specifying their cell fates during cell proliferation. Such a developmental approach results in increasingly committed cells with defined functions to allow for communication and, therefore, maintain homeostasis in their surrounding environment. The underlying molecular mechanism, recognized as the epigenetic machinery, governs such processes in the genome to restrict developmental plasticity during development. The epigenetic machinery manages epigenetic changes through DNA methylation, and chromatin modification is imposed upon gene expression patterns in an inheritable manner. Such a potency restriction in mammals’ differentiated cells has long been considered an irreversible process in vivo. However, recent seminal findings have discovered that such a process could be reversed and reprogrammed in vitro.

A differentiated cell can reacquire pluripotency via three approaches, namely: somatic cell nuclear transplant (SCNT) [6,7,8,9,10,11], heterokaryon cell fusion [12,13] or transcription-factor-induced reprogramming (iPSCs) [14,15,16]. Even though these three reprogramming approaches may use different routes to gain pluripotency, the remodeling of epigenetic configuration still relies on the common themes of re-establishing DNA methylation, chromatin modifications, and transcriptional network formation.

### 2.1. DNA Methylation

As the determination of a cell’s identity relies on its unique epigenetic configuration, faithfully passing down such an epigenetic memory to daughter cells is imperative in order to maintain the cell lineage specification. Therefore, changing the cell fate has to counteract these pre-deposit epigenetic memories. One of the robust epigenetic lock-ons is the high level of CpG methylation constituting 70~80% of somatic tissues [17,18]. Thus, DNA demethylation is one of the essential steps to remodel the epigenetic landscape. Presently, the molecular mechanism of DNA demethylation is recognized to include a passive or active process [19]. Under the condition of lacking functional DNA methylation maintenance machinery, the newly synthesized DNA strand fails to maintain its methylation pattern, as seen in its complement template during DNA replication. Such a process is known as passive DNA demethylation (Figure 2A).

In contrast, active DNA demethylation depends on demethylation enzymes without DNA replication [19]. One of the DNA demethylases is AID (activation-induced cytidine deaminase), found initially to deaminate C to U in the DNA of immunoglobulin genes and to cause somatic mutations or class-switch recombinations. The AID-mediated DNA demethylation process involves deamination of the methylated cytidine residue in single-stranded DNA followed by base excision repair (BER) machinery to replace an unmethylated one [20]. Another DNA demethylase is the Tet family member (Tet1, Tet2, and Tet3), orderly converting 5-hydroxymethylcytosine (5mC) into 5-hydroxymethylcytosine (5hmC), 5-formylcytosine (5fC), and 5-carboxylcytosine (5caC) through a series of oxidation processes (Figure 2B). The final step of removing these oxidative methylcytosines is through the thymine DNA glycosylase (TDG)-mediated excision of 5fc and 5caC. Eventually, the base excision repair (BER) mechanism will participate in the demethylation process (Figure 2B). The overexpression of both AID and Tet1 in highly differentiated cells reshapes the epigenetic landscape [21,22,23]. Although demethylation of DNA is an essential step to overcome the epigenetic barrier, it is of note that extreme demethylation may cause detrimental effects due to genome instability [24,25].

### 2.2. Chromatin Modifications

In addition to depositing unique DNA methylation patterns, shaping the genome during cell differentiation requires a gradual process to install repressive histone marks and increase chromatin compaction [26,27,28]. To re-install a permissive chromatin state, one will erase the repressive histone marks constituting the chromatin-mediated barrier, which is another primary process to gain pluripotency (Figure 2C). One of the prominent repressive histone marks is H3K9me3 as it forms broad heterochromatin domains of chromatin to inhibit transcription factor binding [29]. Thus, H3K9me3-marked regions are considered to be chromatin-mediated epigenetic barriers guarding cell identity [29,30,31]. In this vein, repressing the Suv39H1 and 2 methyltransferases results in H3K9me3 level reduction and facilitates in binding the reprogramming transcription factors [29,32]. In addition to governing a genome-wide heterochromatin domain formation, H3K9me3 can also specifically achieve the silencing of lineage-specific genes by partnering with Setdb1, another H3K9me3 methyltransferase [33,34].

Another repressive histone mark is H3K27me3. Both H3K27me3 and the polycomb complex (PRC2) function together to keep silencing large bodies of development-participated genes in a metazoan genome. Releasing the H3K27me3 marked histones from chromatin leads to a more interactive chromatin state and yields a transient primed chromatin state to facilitate cell fate change [35,36,37]. Utx is a JmjC domain-containing enzyme that mediates the H3K27me2/3 demethylation process [38]. Consistent with its role, the depletion of Utx significantly reduced the reprogramming efficiency in mouse embryonic fibroblasts [39].

### 2.3. Transcriptional Network Formation

Early days’ studies demonstrated that cell lineage can be defined by transcription factors; the expression of the master regulator can re-define a cell’s identity. For example, the ectopic expression of MyoD alone in fibroblasts and adipoblasts is sufficient to change into myoblasts [40]. Likewise, the overexpression of C/EBPa allows for lineage conversion from B cells to macrophages [41]. Moreover, the Ngn3, Pdx1, and MafA gene cocktail convert pancreatic exocrine cells into beta-cells [42]. Thus, master regulatory genes or gene sets organize unique transcriptional network formation as a prerequisite for establishing the corresponding cell identities. In this vein, the reprogramming process of the cell fate will need to progressively cease the existed transcriptional networks, while being relayed to other transcriptional programs for re-establishing a specific cell type of interest.

As mentioned previously, it is intriguing to know how several transcription factors function together so as to define their unique cell identity. Graf and Enver proposed a transcription factor cross-antagonism model to address cell fate determination and transition during reprogramming, including the events of de- and trans-differentiation [43]. This view was further supported after uncovering the mechanism of pluripotency acquisition during somatic programming. The traditional core pluripotency factors, including Oct4, Sox2, Tbx3, and Nanog, are recognized as members of the germ layer specifiers. They function in a manner of precarious balance through antagonizing and cross-activating each other to reach a pluripotent state. The antagonistic effect halts the pluripotent state from falling into any germ layer lineages [44]. This idea was further fortified in the cell fate change through the overexpression or knockdown of those lineage specifiers. For example, the knockdown of Oct4 in ESC renders it incapabile of mesoderm formation [45]. Likewise, compromised Nanog leads to the failure of mesendoderm differentiation [46]. Therefore, one can surmise that Oct4, Nanog, Sox2, and Tbx3 dictate lineage differentiation in ESCs and maintain ESC’s pluripotency through antagonizing each other.

### 2.4. A Two-Way Relationship between Transcription Factor and Chromatin Structure

It is neccessary to note whether epigenetic alternation is a prerequisite to layout an adequate chromatin microenvironment for the newly established transcriptional networks to work on. Alternatively, the transcriptional complex may actively modify the chromatin configuration to build a permissive domain for the subsequent cell identity establishment. Some transcription complexes contain chromatin modifiers and use their specific DNA binding domain to bring the associated modifiers in order to achieve an active, repressive, or bivalent chromatin configuration in their binding vicinity [47,48] (Figure 2C). There is mounting evidence to support both views. Interestingly, the recent chemical reprogramming approach without engaging an ectopic factor-driven force could also achieve cell fate changes. These findings indicate that the transcription network and chromatin modification are both effective for achieving a synergistic reprogramming effect [49,50,51,52]. Therefore, the current view of a two-way relationship between transcription factor binding and the chromatin structure has been proposed during cell fate reprogramming.

### 2.5. Diverse Mechanisms Coordinating on Genome Plasticity

The epigenetic configuration defines cell fate, creates population heterogeneity, and governs a differentiation-priming event. The initiation of epigenetic modifications may derive from “symmetry-breaking”, and subsequently result in the differentiation priming [53]. The created hemimethylated sites in the newly synthesized daughter DNA strand may not be synchronized and maintained by the UHRF1/DNMT1 complex during cell proliferation. Thus, it creates a transient time window in order to allow other signals to diversify the two daughter cells’ fates further. The heterogeneous pattern of DNA methylation generated within that brief moment enables an independent differentiation-priming between two daughter cells before their commitment. Likewise, the dynamic bivalent chromatin structure on the developmentally regulated genes potentially adds another layer of “entropy” to create cell fate diversity. The cross-regulatory mechanism between transcription factors acting together with the chromatin structure and DNA methylation levels ensures that the precise signal interpretation from its dwelled surroundings guides the stereotype developmental path within a species.

## 3. A Transgene-Based Pluripotency Acquisition

The initial observation of the cell fate change was mainly from one cell lineage to another, indicating the plasticity property of the mammal genome. An observation of the dramatic cell fate change was not made until performing somatic cell nuclear transplant (SCNT) [6,7,8] and heterokaryon experiments [12,13], where a highly differentiated nucleus was reprogrammed to a pluripotent state. Furthermore, a transgene-mediated approach, namely the induced pluripotent stem cell (iPSC), was followed to achieve an ESC-like pluripotent state from highly differentiated somatic cells [14,15,54]. Although SCNT and heterokaryon experiments were performed much earlier than for iPSC, their pluripotency re-acquisition process and the corresponding mechanisms were revealed from recent studies [55]. Here, we only focused on the current progress in unraveling the transgene-based reprogramming mechanism, even though it is very intriguing that for both SCNT and heterokaryon, the change in chromatin assembling is independent of DNA replication. An excellent review of the recent progress in SCNT can be found elsewhere [55].

In 2006, a pioneering work lead by Takahashi and Yamanaka demonstrated that a highly differentiated somatic epigenome could be converted into an ESC-like pluripotent epigenome with only four transcription factors, namely Oct4, Sox2, cMyc, and Klf4. This groundbreaking work reified our current concept of genome plasticity. Regarding the efficiency of reprogramming in the original Yamanaka study, it displayed only approximately 0.001~0.01%, which was close to the estimated abundance of the residing stem cells in a tissue [14]. Because of the low frequency of iPSC formation in the first report, the central question focused on what mechanism impeded epigenome remodeling. As reprogramming turns the developmental clock backward, it is conceivable that the machinery of the cell state maintenance will be the first encountered barricades regarding epigenome landscape remodeling from a highly differentiated status to a pluripotent configuration. Following this, Cheloufi et al. performed two RNAi screens, where the histone chaperone CAF-1 was identified during transgene-mediated iPSC formation. Suppression of CAF-1 facilitates chromatin structure accessibility at enhancer elements by reducing heterochromatin domains during early reprogramming. In addition, RNAi CAF-1 enhances the direct lineage-crossing of B cells into macrophages and that of fibroblasts into neurons. Thus, the histone chaperone CAF-1 represents the safeguard of somatic cell identity [1].

As most highly differentiated somatic cells exit the cell cycle, they encounter a powerful mechanism to “lock on” a differentiated epigenome, as gaining pluripotency during iPSC formation requires cell cycle re-entry to restate the chromatin configuration in a potentiated bivalent fashion. After Yamanaka’s seminal finding, several groups reported that reduced or eliminated tumor suppressor genes, such as p53 and Ink4a/Arf, accelerated cell division and led to a higher iPSC production. As the p53-p21 pathway guards cell cycle progression, it represents a roadblock during cellular reprogramming [56,57,58,59]. Additional factors like Glis1, SUV39H1, DOT1L, and YY1 were also shown to alter the iPSC production efficacy [32,60]. Identifying the endogenous Essrb and Utf1 expression further served as a predicter for successful reprogramming [61]. Collectively, these underlying mechanisms of epigenome plasticity mentioned above (i.e., DNA methylation and chromatin reorganization and transcriptional network formation) sketched the outline of the current reprogramming concept. Although they may take diverse routes, reprogramming events act in an ordered and concerted multiple-step manner. The following apparent questions have been raised at each transit phase: How do highly differentiated cells decide to initiate the reprogramming process? What is the roadblock impeding iPSC formation at each stage? What are the players participating in the dynamic epigenome conversion at each step? What is the safe and efficient way to generate iPSC for clinical uses? The following sections may shed light on potential answers.

## 4. The Route Choice—Molecular Control of Induced Pluripotency Initiation

Even though the dynamic molecular profiling on different stages/phases of reprogramming has been uncovered in the past decade (discussed in the next section), the onset of significant participants and its route-taking decision have remained elusive. Reprogramming route choices are various depending on the cell types, fates, and the iPSC-generated approaches. Several reprogramming modes of action have been observed, i.e., the stochastic, deterministic, early stochastic, late deterministic, and biphasic models [62,63]. These action modes can be distinguished by the timing of the phase progression and the analytic approach adopted during the epigenetic landscape’s remodeling (Figure 3A). For example, the stochastic model states that all candidate cells after certain cell divisions have an equal chance of reprogramming. The acquisition of pluripotency, therefore, is a random event. Such a reprogramming event is primarily described as using the mouse embryonic fibroblast (MEF) as the initial cell source and is driven by the OSKM transgene for reprogramming [63]. Under the OSKM-mediated transgenesis, additional genetic manipulations are amenable in order to accelerate the kinetics. For example, Hanna et al. observed kinetic increment in the clonal analysis of single B cells by either enhancing cell proliferation or the cell-intrinsic mechanisms [2].

Despite the prevalence of the stochastic action mode, the deterministic model has also been observed in a subpopulation of fast-cycling bone marrow cells. Apart from the stochastic model, only specific cells with a privileged state can overcome epigenetic barriers during the deterministic process. Therefore, it is synchronized to surpass the encountered barrier’s hurdle, e.g., the mesenchymal-to-epithelial transition (MET), while adopting fibroblast as a reprogramming cell source [65,68]. Furthermore, to facilitate the deterministic route, C/EBPα can be overexpressed or Mbd3/NurD can be depleated through genetic manipulations [65,66,68]. Although each mode of action has strong supportive evidence, the molecular mechanisms regarding the process of each mode entry at the onset of reprogramming and the subsequent route choice are still mostly unknown. Recently, Liu and colleagues tackled this issue by observing a convertible stochastic and deterministic process while adopting the MEF from the Col1a1 4F2A Oct4-GFP triple transgenic mouse system [67]. Importantly, they found that the crosstalk between the prior recognized LIFR/GP130-STAT3 and EGFR/Erb2-Erk1/2 signal transduction pathways made the route decision (Figure 3B). As they used the extracellular domain of E-cadherin (NTF1) to serve as a signaling molecule for modulating both of the paths mentioned above, the final route-to-be-taken decision depends on the candidate MEFs according to the output of the LIFR/GP130-STAT3 and EGFR/Erb2-Erk1/2 pathways. Thus, under the scenario of typical OSKM expression, the stochastic route will be taken preferentially. In contrast, the deterministic process will be adopted in the presence of the extraneous E-cadherin recombinant protein (NTF1).

Accordingly, the entry mode selection at the onset of somatic reprogramming mainly depends on the differential outputs of pSTAT3 and pErk1/2 influenced by its residing microenvironment, if they are not the only determinant factors. Intriguingly, this is the same pathway functioning at the late reprogramming transition states [69,70,71]. Thus, by manipulating both pathway outputs, one can leverage the action mode toward one or the other, the stochastic or the deterministic, and vice versa [67]. In this vein, the mode choice will depend on that particular moment of cell state at the reprogramming onset, which is affected mainly by the microenvironment. Thus, context-specific eliteness can also be a deterministic element contributing to cellular reprogramming by dominating the reprogramming niche. As Liu’s results depicted that neither route choice increases the proliferation rate nor affects the reprogramming kinetic, different MEF subpopulations may likely participate in different reprogramming modes. It would be interesting to see whether the cells gaining pluripotency are the same subpopulation under different action modes and whether other cell types besides MEFs warrant the same rule.

Recently, Shakiba et al. combined cell-barcoding and lineage-tracing strategies to demonstrate that the resultant iPS clones arise from the poised MEF subpopulation with a Wnt1-expression [72]. Moreover, such Wnt1-bearing cells, presumably representing a neural crest population, dominate to gain the reprogramming niche [72]. Like MEF, reprogramming human somatic cells to pluripotency remains inefficient due to the lack of mechanistic understanding. It has been reported that a two-phase process is observed in human somatic reprogramming, i.e., a prolonged stochastic phase followed by a rapid deterministic phase. In this vein, the early stochastic process is the rate-limiting step governing the success of gaining pluripotency. To unravel the early stochastic mechanism, Chung and colleagues adopted the approach of single-cell transcript profiling along with mathematical modeling to demonstrate that the stochastic phase is an ordered probabilistic process with independent gene-specific dynamics [73]. Furthermore, their results indicated that chromatin modifiers could be used to discern whether reprogramming cells are on a successful reprogramming trajectory. Thus, human somatic reprogramming is consistent with the notion above in mice, in that chromatin remodeling plays a pivotal role in somatic reprogramming [73]. As the onset of the route choice serves as a significant factor in the reprogramming efficiency, understanding such vital events and following reprogramming phases will give better insight into the reprogramming process at the molecular level.

## 5. The Routes—Stepwise Phases of Reprogramming Mechanism

### 5.1. The Pros and Cons of Different Transgenesis Systems Used in iPSC Production

As the robust expression of Yamanaka factors warrants the success of pluripotency acquisition, the retrovirus-mediated transgenesis system serves as a routine basis for iPSC production in most laboratories due to its high infection rate and robust transgene expression. However, the action of retrovirus-based transgenesis results in transgene insertions, which may create unwanted mutations and affect genome stability. Although episomal vector-, chemical-, protein-, and mRNA-based reprogramming strategies alone avoid the insertion of transgenes, the reprogramming efficiency is low and may require multiple Yamanaka factor introductions during the process of induction [14,74,75,76,77,78]. Following the awakening of Sleeping Beauty, this ignites the hope for advancing non-viral-based gene therapy to routine clinical applications and using mammalian transgenesis for functional genomics [79,80,81,82]. Our pioneering works have also demonstrated that piggyBac is a flexible and highly active transposon compared to Sleeping Beauty, Tol2, and Mos1 in mammalian cells [83,84]. Nagy’s lab first adopted piggyBac transposon and demonstrated the transgene-free iPSC production in mice [85,86]. Furthermore, Vallier and colleagues adopted the piggyBac traceless removal property to successfully correct alpha1-antitrypsin deficiency in mouse iPSC [87]. To avoid the mutagenesis occurrence derived from transgene insertion, several episomal plasmid and viral systems have developed; for example, the plasmid carried a human Epstein−Barr virus (EBV)replication origin [88] or the Sendai virus [89,90]. Thus, one could have a sufficient Yamanaka factor expression and remove the transgene after the cell acquires its pluripotency.

Another concern regarding the efficiency and completeness of reprogramming is the dosage of each driving factor and their combinations. As the developmental potential of iPSCs is greatly influenced by reprogramming factor selection, one major determinant of iPSC quality relies on the variety of reprogramming factors used [91]. To avoid the complicated genetic background in humans, Churko and colleagues generated hiPSC from the same human primary fibroblast using six different reprogramming methods as mentioned above [92]. Although all of the hiPSC lines produced from different approaches could proceed to differentiation, the resulting transcriptomes are displayed differently due to various epigenetic signatures [92]. Further studies on other isogenic fetal organs as reprogramming cell sources have demonstrated that all tissue-specific DNA methylation patterns might confer a comparable DNA methylation configuration as the existing hiPSC lines [93]. However, brain-specific DNA still preserved its epigenetic memory, leading to a higher propensity to differentiate to its neural origin [93]. Thus, tissue-specific DNA methylation patterns might affect the completeness of reprogramming [93].

Furthermore, the stoichiometry of reprogramming factors also played an essential role in reprogramming [94]. For example, combining a higher expression of Oct4 and Klf4 with a lower c-Myc and Sox2 generated all-iPSC mice efficiently because the imprinting at the DLK1-Dio2 locus was typically maintained. Although the loss of imprint at the DLK1-Dio2 locus still produced iPSC, the all-iPSC mouse cannot be efficiently made [94]. Thus, the stoichiometry of the reprogramming factors governed the quality of iPSC by preserving its proper imprinting at the DLK1-Dio2 locus [94]. Despite the advances in assembling the ideal factor set for obtaining high-quality iPSC, various studies have reported the accumulation of abnormalities in the reprogrammed genome. These genetic and epigenetic aberrations are well documented, including DNA methylation pattern alteration, parental imprinting, and X chromosome inaction [95]. Therefore, advancing the transgenesis system with adequate reprogramming factor combinations and cell sources are a future direction for iPSC’s clinical applications.

### 5.2. The Transgene-Based Somatic Reprogramming

Most of the reprogramming experiments have done using the transgene-based strategy. Herein, we focus on delineating the transgene-based mechanism of reprogramming in a stepwise manner. Several efforts, including genome-wide analyses of the transcriptome, proteomics, metabolism, and epigenetics, have focused and attempted to unravel the somatic reprogramming mechanism by identifying its stage-dependent landmarks [5,96,97,98,99,100]. The transcription profile of those tested transgene-based iPSC (tgiPSC) lines displays a high similarity to ESC and a considerable contrast to its tissue origin. However, Daley and colleagues found different tissue-derived iPSCs harbor residual DNA methylation signatures resembling the pattern of their somatic tissue origin [101]. Such an iPSC with a residual epigenetic memory of the donor tissue facilitates its differentiation to the original cell type rather than the other unrelated lineages. In contrast, SCNT displays pluripotency ground state establishment more readily than factor-based iPSC reprogramming [101]. Hence, the phenomenon of epigenetic memory is mainly due to the incompleteness of somatic reprogramming. To explore potential markers indicating a complete epigenetic memory erasure, Hochedlinger and colleagues verified that the imprinted *Gtl2* gene expression should ensure all-iPSC mouse production via tetraploid complementation. Such findings are consistent with the developmental roles of the Dlk1–Dio3 gene cluster [99]. Furthermore, ascorbic acid was shown to preserve the imprinting status of the Dlk1-Dio3 locus and to improve iPSC efficiency, while it was supplemented in a culture medium [102,103]. Altogether, a tissue’s origin can strongly influence the epigenetic status and biological properties of the resultant iPSCs. The presence of vitamin C during the reprogramming process substantially improves both the efficiency and quality. The effect of vitamin C is thought to act as TET’s cofactor modulating its enzyme activity through promoting TET folding and FE(II) recycling [104,105,106].

As reprogramming is an asynchronous process containing a heterogeneous population, the bulk population-based approaches impeded the sequential and timely dissection. Such an asynchronous and heterogeneous nature mainly derives from the Yamanaka factors’ expression dosage or the early reprogramming route choice, which are not mutually exclusive [2,62]. Therefore, gaining temporal control of the transgenes’ expression and generating a “reprogrammable system” for iPS-transgenics will vastly ameliorate such issues [99,107,108]. Two labs applied this regulatable iPS-transgenic mouse and reported an in-depth analysis on dynamic profiling of proteome and transcriptome of reprogramming MEFs into iPSCs [5,109]. After analyzing approximately 8000 proteins, Hansson and colleagues uncovered a two-step resetting of the proteome during the first and last three days of reprogramming over a two-week interval, while more subtle changes occurred in the intermediate phase [5].

Furthermore, the biphasic protein expression profile derived from the ontology analysis displays a highly coordinated fashion with functionally related proteins [5]. Coincidentally, Hochedlinger and colleagues applied the same system to perform genome-wide transcriptome analysis, and witnessed that two transcriptional waves were concomitantly expressed along with the Yamanaka factors during reprogramming [109]. Apart from the observations from the sing-cell expression analyses, their data suggest that iPSC formation follows an early and late deterministic phase separated by a more probabilistic phase [61]. The action mode inconsistency observed in cellular reprogramming may reflect the nature difference in population- *vs.* single cell-based analyses. Nevertheless, the point of agreement from those studies demonstrated that reprogramming is a stepwise event through tightly temporal control of those functionally related molecular networks.

Identifying the different factors that participate in the various stages will improve the efficiency and completeness of the somatic reprogramming. In addition, an orderly reprogramming process helps identify distinct hierarchy regulatory networks governing each different reprogramming phase [61,110,111,112]. Most of the reprogramming studies adopted MEFs as the initial cell source. Based on the MEF-mediated iPSC formation, the current consensus of the primarily reprogramming process constitutes three essential phases: initiation, maturation, and stabilization (Figure 4, left panel). The molecular event at each stage was addressed as follows.

### 5.3. The Initiation Phase

Based on the extensive transcriptome profiling of those reprogramming fibroblasts in the bulk culture, the loss of fibroblast identity (e.g., a highly expressed surface antigen Thy1 in fibroblasts) accompanied with cell morphology alternation was observed at this stage by engaging the mesenchymal-to-epithelial transition (MET). Such a signature is characterized by losing the expression of Snail1/2 and Zeb1/2 and gaining the epithelial makers [3]. The induction of miRNA-200 family members, Epcam, and Cdh1 indicates a cell fate change through waning the TGFbeta pathway [110,116,117]. One of the critical events underscores the needs of the Tet1, 2, and 3 activities during the initiation phase of chromatin remodeling as activation of the miR-200 family requires Tets [118]. After overcoming the barrier of MET, reprogramming cells acquire proliferation capability and refuted apoptosis, allowing for replication-mediated epigenetic modification [56,58,59]. During the MET process, the loss of Thy1 and CD44, accompanying the gain of alkaline phosphatase and SSEA1, are typical changes observed in MEF-based reprogramming [3,109,110]. As various routes may take place at the beginning of reprogramming, gaining the expression of SSEA1 does not guarantee the SSEA1+ cells will eventually become iPSCs at this stage. In general, upregulation of the genes related to cell proliferation, metabolism, and cytoskeletal organization is observed at this stage, whereas development-related genes are mainly downregulated through histone modifications. Intriguingly, DNA methylation rearrangement does not occur at this moment until a later phase of reprogramming [109]. Although this initial process may follows various routes, it does not seem to hurt the process as long as it meets all the initiation phase’s modifications [61].

### 5.4. The Maturation Phase

The onset of the maturation phase is characterized by the pluripotency genes’ expression, especially Oct4, Fbxo15, Nanog, and Sall4 [5,109,110]. Most importantly, only a subset of pluripotency genes, but not all, is associated with the maturation phase. This phenomenon indicates that a sequential event must be strictly followed during the progression of this phase. However, again, the appearance of the pluripotent markers mentioned above has not yet assured the completeness of somatic reprogramming [61,109].

Another essential feature of the late reprogramming event is transgenes’ silence. Such a phenomenon indicates the cells are situated in a transit-competent state and are ready to enter the next stabilization phase [110]. A crucial regulatory network has been identified to survive these reprogramming candidates after transgene suppression. Again, at this stage, even the presence of Oct4 or Nanog, assuring self-renewal independent from transgenes, does not warrant successful reprogramming. The function of such a regulatory network mainly engages a subset of pluripotency-associated factors from a suppressive state to a poised manner via alternation of the DNA methylation pattern [109]. Thus, the maturation phase involves a time-consuming molecular reactivation event. The recruitment of polycomb group and NuRD complex may partially explain such a stepwise factor-activating event at this unique stage [30,39,68,119]. However, how such a sequentially activating process compensates for the loss of the Yamanaka factors to stabilize the maturation state remains elusive.

### 5.5. The Stabilization Phase

The transit from the maturation to the stabilization phase will acquire additional pluripotent factors, such as Utf1, Lin28a, Dppa2, and Dppa4, to relay the pluripotent state after losing the transgene [3,96,120,121,122]. Meanwhile, numerous epigenetic alternations are concomitantly accomplished. These changes, including restoring telomere length, reactivating the X chromosome, and resetting the cell type-dependent residual epigenetic memory, are primarily achieved through DNA methylation rearrangement [3,56,123]. This progressive DNA methylation reconfiguration starts from the late onset of maturation and spreads throughout the entire stabilization phase. Several participants, for example, AID, TET family, and DMNTs, are responsible for epigenetic landscape remodeling at this stage [109].

### 5.6. The Chemical-Based Somatic Reprogramming

The reprogramming mechanism mentioned above is seen from the transgene-based somatic reprogramming. Recently, Deng’s group successfully developed a chemical-based reprogramming approach to produce an integration-free and oncogene-free pluripotent cell in humans and mice, namely chemically induced pluripotent stem cells (CiPSC) [49,124]. As can be expected, alternative approaches may result in different route choices during reprogramming (Figure 4, right panel). For example, a sequential XEN-like intermediate (i.e., Gata4, Gata6, Sall4, and Sox17) followed by the two-cell-like state (2C-like) (i.e., Zscan4, Tcstv1, Dppa, and Oct4) was reported before reaching the bona fide pluripotency in CiPSC [50,52,124,125,126]. However, CiPSC takes somewhat different routes from the OSKM transgene-mediated reprogramming process, transiently expressing the 2C-like gene set, of note, which is the most intriguing part of this reprogramming process. Several transgene-based studies also observed a transient entry of the 2C-like state [113,114,115]. The progression of reprogramming creates high heterogeneity, impeding the identification of a small fraction of cells in the iPSC-forming course through the traditional bulk RNA-profiling. Dissecting such sophisticated biological processes will need a single-cell resolution to capture such rare events. Perhaps for this reason, the earlier engagement of bulk transcriptome analyses failed to identify the subpopulation situated in such a state (Figure 4, right panel).

Alternatively, trespassing the 2C-like state may reflect a nonspecific event created by an unstable epigenome during the genome-wide demethylation process at the late stage of reprogramming. Regarding this issue, Deng’s group adopted a high-resolution scRNA-seq to dissect the process of CiPSC from MEF. The reprogramming process was arranged through a pseudo-timing analysis into three stages (i.e., stage I, II, and III). The 2C-like program falls into stage II, accompanying the occurrence of global hypomethylation. Those upregulated genes in the 2C-like stage, including Dppa2, Dppa4, Klf2, Zscan4, Gm13154, and Tcstv, were reminiscent of a two-cell embryo’s expression profile. Furthermore, substantially enhancing the 2C-like program accelerates CiPSC formation, verifying that the 2C-like program indeed serves as a driving factor of the late transition [52].

Recently, the observed metastable states of ESC shuttles reversibly between the 2C-like and pluripotent state, providing a unique mechanism in maintaining the genome integrity of ESC in vitro [127]. Because of the 2C stage-specific expression of Eif1a inhibiting DNMT protein synthesis, the resultant genome-wide hypomethylation at the 2C-like stage may facilitate epigenome-resetting and, in turn, endow the following naïve pluripotency establishment [128,129]. Although a pluripotent state can be achieved by different route choices (tgiPSC vs. CiPSC), the ordered phases during each reprogramming progression can be unambiguously discerned. The agreement point of the alternative route-taken, as mentioned above, is that relaying a unique transcript circuitry to another designated transcript network is the way to accomplish cell fate transition.

### 5.7. The Distinctive and Common Phases between tgiPSC and CiPSC

The formation of both tgiPSC and CiPSC are all temporally organized multistep processes, where the sequential epigenome conversion results from linking the distinct molecular networks [111]. In the sense of tgiPSC, an existing single master regulatory gene, like MyoD for myogenesis, is unlikely to be the case at each step of stemness-gaining. More likely, each step (i.e., initiation (MEF), MET, pre-iPSC, maturation, and stabilization (iPSC)) may contain molecular circuitry to stabilize that individual cell state with a corresponding epigenomic configuration. The transit between phases will coordinate the changes of each stage-specific molecular network along with its epigenome conversion. Hence, the progressive relay in each step toward pluripotency is expected to encounter the stage-specific barrier and needs to be rewired from one molecular network to another state-specific circuitry. Accordingly, leveraging each stage-specific molecular network by overexpressing its primary player(s) will uplift the molecular network in a transit phase for the subsequent network re-arrangement, while proceeding toward pluripotency.

In the case of CiPSC, the transit of cell states is not driven by the ectopic expression of the transcription factor set. Instead, it promotes or represses specific pathways at each step through different small molecular cocktails in order to create a permissive chromatin configuration. Consistent with the two-way relationship, exerting a mutual effect to promote reprogramming by transcription factors and chromatin modifiers, a 16–24 day prolonged process is required for CiPSC formation [52]. A distinctive difference between tgiPSC and CiPSC is the route choice, where CiPSC first passes the XEN-like state but not MET. The subsequent occurrence of global DNA hypomethylation resulting from the Ci2C-like state entry is required in order to achieve its final pluripotent state. Although some tgiPSC studies have reported that the 2C-like state is also needed, as mentioned above, the disagreement is still unsettled [113,114,115]. Nevertheless, Dppa2 and Klf2, the significant components in the 2C-like state, appear in the tgiPSC stabilization phase and may correspond with this possibility. Of note, the Ci2C-like state, 2C-like state, and 2C embryo all share similar expression profiles, but neither are identical. Understanding molecular network rewiring during each transitional stage in different approaches will shed light on how the plasticity of the genome is established and how efficient, precise reprogramming can be achieved.

## 6. Reprogramming beyond Pluripotency

### 6.1. The 2C-like State

While unraveling the pluripotency acquisition mechanisms and route choices, several attempts have concomitantly focused on further surpassing the pluripotency boundary to reach a totipotency-like summit state of a differentiation capability. In 2012, Macfarlan and colleagues reported a rare transient population within ESC clones presenting two-cell-like (2CLC) features in the serum/LIF culture system. As 2CLC was named after its shared features with the two-cell stage embryo, such an expended ESC’s plasticity can readily serve as an in vitro totipotency model [130,131]. More supporting evidence from the in vivo reprogramming experiment observed that the generated iPSCs acquire their totipotency features besides producing a three-germ layer in the derived teratomas [132]. These cells, however, cannot be stably maintained in an in vitro culture system, as it failed to test their developmental potential as totipotent cells. Thus, such an oscillating transit of cell fate between the 2CLC totipotent-like to the pluripotent state reflects the failure of the current culture system to capture such a unique 2CLC state.

### 6.2. Expanded Potential Stem Cell—An In Vitro Captured 2C-like State

In 2017, two groups independently established chemical cocktails for those extended/expanded pluripotent stem cells (collectively known as EPS cells (EPSCs)), which allowed for their stable long-term maintenance and conferred a totipotency-like competency in mESC, hESC, and the human fibroblast-derived iPSC [133,134]. Interestingly, even though those EPSCs displayed an increased developmental potential, their global gene expression pattern varied from two- to four-cell morula stage embryos. Therefore, those EPS cells have a unique cell state similar to, but distinct from, 2CLCs and 2C embryos. Of interest, although both 2CLCs and EPSCs can form extraembryonic tissues in chimeras, as their names suggest, 2CLCs, in contrast to EPSCs, do not express extraembryonic markers [135,136].

As mentioned above, hiPSC can be used to advance to the totipotent state under those unique cultural conditions. However, it has not been shown that is can reach totipotency directly from a highly differentiated fibroblast cell in one step. Further modifying the transgene combinations may be possible as it has been demonstrated that overexpressing a single Axin gene in mESC can obtain mEPSC in EPSCM (the medium containing the cocktail of inhibitors and LIF for the expanded potential stem cell) [133]. Despite these mice and human EPSCs contributing to the extraembryonic structure in chimeric mouse embryos, individual EPSC is unlikely to produce an animal independently. Thus, the above-defined “extended” or “expanded” potency only displays certain traits of totipotency, apart from the traditional definition of totipotency in development. In line with pluripotency’s historical term, Redó Riveiro et al. have suggested the term “experimental totipotency” to address a cell’s differentiation capacity, contributing to both embryonic and extraembryonic tissues in a chimera experiment [137].

## 7. Reprogramming to Generate Germ Cell

A zygote, the fertilized egg, is a genuine totipotent cell capable of forming an entire organism independently [138]. After fertilization, the sperm pronucleus is triggered to reprogram through the maternal factors in the ooplasm. Similarly, both SCNT and heterokaryon exposed their differentiated nuclei in the ooplasm or ESC’s cytosol, where plasma factors initiated the process of reprogramming. However, somatic reprogramming to reach such zygotic totipotency may have difficulties, because the reprogrammed cell hardly gets the cell size, epigenomic landscape, and maternal contents as seen in a zygote or a blastomere of the pre-implantation embryo. Reprogramming to generate the male and female gametes is thus an alternative way to reach totipotency.

As germ cells play a role in linking generations and are a mediator of evolutionary forces in natural selection, their epigenetic organization plays an essential role in perpetuating an organism’s life cycle and representing its adaptive outcome. During early development, primordial germ cells (PGCs), the founder cells of the sperm and egg, are specified and relocated to the gonad to acquire further maturation instructions. Despite the existing differences between mouse and human germlines regarding their origin, genetic networks, and maturation phases, the process of mouse gametogenesis still serves as a valuable, informative model to compliment the unavailability of human counterparts [139,140].

Understanding the timing of the primordial germ cell’s (PGC) presence and the surrounding microenvironment inputs during embryogenesis will help reconstitute germ cell development in vitro using PSCs as the starting materials. As such, both male and female mouse primordial germ cell-like cells (mPGCLCs) were induced from Epiblast-like cells (EpiLCs) and gave rise to fertile offspring after transplanting into the testis and under the ovarian bursa, respectively, of an immunodeficient mouse [141,142]. Furthermore, mPGCLCs were successfully differentiated into primary oocytes at the secondary follicle stage while co-cultured with embryonic ovarian somatic cells (the reconstituted ovaries (rOvaries)) [143]. Similarly, in vitro spermatogenesis was generated from reconstituted testis (rTestis) from mPGCLCs [144]. Those reconstituted gonad culture systems further provide adequate niche support to produce fertile offspring.

Recently, a xenogeneic rOvary was established by aggregating hiPSC-derived hPGCLCs with mouse embryonic ovarian somatic cells. Over 3~4 months of co-culture, hPGCLCs underwent genome-wide DNA demethylation and were differentiated into retinoic acid-responsive human fetal germ cells (FGCs). FGC is an immediately precursory state for meiotic oocytes, equivalent to the 12th week of human development. However, FGCs failed to enter meiosis to complete their differentiation further [145]. The most recent study showed that hPGCLC co-cultured with the mouse embryonic xenogeneic rTestis and was differentiated into prospermatogonia [146]. Nonetheless, it is critical to scrutinize the potential genome mutations and epigenetic abnormalities during each in vitro process of amplification and differentiation [143,144]. Thus, following the developmental track of fertilization through the iPSC-derived sperm and egg, one can now reach a bona fide totipotency in the mouse embryo, which can be expected in humans soon.

## 8. The iPSC-Based Disease Modelling and iPSC Therapy in Clinical Trials

Since iPSC provides an unprecedented opportunity to model diseases, reprogramming technology has been adopted to study the etiology of previously inaccessible diseases. Due to the incomplete removal of residual epigenetic memory, it tends to differentiate toward its original cell type. The iPSC-derived phenotypic cell often exhibits immature functional characteristics, a reminiscence of its respective embryonic or fetal phenotypic cells, and accompanies a heterogeneous population. Thus, iPSC-mediated disease modeling has been applied more adequately for those early-onset diseases, but less successfully in late-onset ones due to lacking the adult maturation characteristics. For example, both the early-onset disease modeling of long QT syndrome and spinal muscular atrophy have been successfully established [147,148].

Of note, the process of somatic reprogramming may accumulate genetic aberrant, including chromosomal abnormalities, copy number variants, and genetic instability during a prolonged culture time, which is needed for epigenetic re-configuration and expansion. Furthermore, avoiding the potential tumorigenicity resulting from incomplete differentiation induction as well as heterogeneity, one, therefore, will need to seriously consider if the cells will be used for cellular transplantation; iPSCs are capable of forming teratomas and malignant tumors, such as neuroblastoma and follicular carcinoma, if the undifferentiated pluripotent state was introduced into the patient [121] (Okita et al., 2007). Thus, strict guidelines will need to define the acceptable quality of iPSCs and their differentiated derivatives for safe and effective clinical applications.

Another practical concern will be the cost of the customized iPSC-mediated therapy, as it has been estimated to cost approximately USD 800,000 to have a tailored clinical-grade iPSC [149,150,151]. To use iPSC-mediated cell therapy, one may consider using allogeneic iPSC-derived cell sources via cell banking instead of adopting an autologous iPSC therapeutic strategy. It will be much approachable to establish a limited number of approved iPSC lines with various human leukocyte antigen (HLA)-homozygous donors. These lines can be subject to regulatory clearance via vigorous tests on their genome stability, viral contamination, differentiation capability, and tumorigenicity [149,150]. Thus, the idea of haplobanking of HLA-homozygous iPSCs was spawned [152,153,154]. Notably, the first clinical trial led by Dr. Masayo Takahashi used autologous iPSC for treating neovascular age-related macular degeneration (AMD). This novel therapeutic modality transplanted the iPSC-derived retinal pigment epithelial (RPE) cells [155]. Two patients were admitted to this first-ever iPSC clinical trial. However, the second patient did not obtain the cell replacement trial due to the identification of genetic alternations in both iPSCs and iPSC-derived RPE cells. One year after surgery, the visual acuity had not improved or worsened in the first patient, even though the transplanted sheet remained intact. Although the trial did not display a beneficial effect, it demonstrated the introduction of iPSC-derived RPE cells did not raise a deterioration effect.

## 9. Conclusions

Along with unraveling reprogramming’s mechanism and advancing versatile iPSC technologies, the potency of the reprogrammed cell has been further extended in recent years. Besides being a three-germ-layer differentiation capacitating cell, one can reprogram somatic cells beyond pluripotency by generating the EPS cell types to reach the experimental totipotency state [133,134]. Although the experimental totipotency can generate the extraembryonic cell types, those cells (i.e., 2CLC and EPSC) cannot form an organism independently as a zygote. Germ cells and the followed early preimplantation embryos after fertilization are the genuine totipotent cells bridging the generations for species propagation. The recent excitement on coaxing mESC and miPSC generates female and male gametes in vitro, and subsequently, producing fertile offspring is a pivotal breakthrough in stem cell and reproductive biology. Although the xenogeneic rOvary system made human FGCs from the hiPSC-derived hPGCLC, FGCs failed to enter meiosis to get maturation further [145]. One could expect such a problem will be solved shortly. In that aspect, the related ethical issues and regulations will have to catch up with the fast pace of such scientific advances.

Since 2014, a burgeoning number of iPSC clinical trials have been launched to test their therapeutic effects in different diseases. The first clinical trial, led by Dr. Masayo Takahashi, RIKEN center in Kobe, Japan, involved transplanting iPSC-derived RPE cell sheets into macular degeneration patients. Following the trial, several BioTech companies also initiated trials for their iPSC-derived therapeutic products across multiple therapeutic areas, including CAR-T, COVID-19, Parkinson’s disease, etc. (Global induced pluripotent stem cell (iPS Cell) industry report 2021). Although the expected therapeutic effect has not been reported after reaching clinical trials’ endpoint, regenerative medicine is believed to be the future trend at the bedside. Furthermore, as somatic reprogramming is one of the rapid-pacing fields in modern biology, understanding the current concepts and correspondingly supported evidence in this field will be instrumental in tackling the emergent issues or difficulties encountered in the future.

## Figures and Tables

**Figure 1 cells-10-02888-f001:**
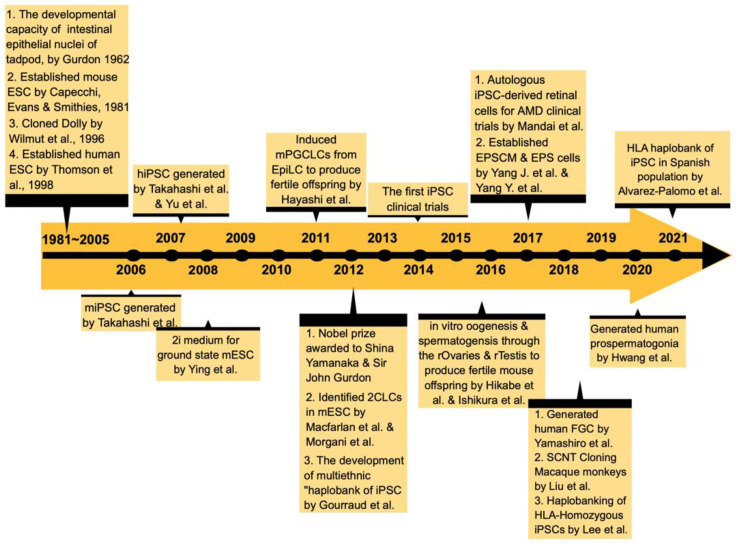
The timeline of significant scientific advances in the history of nuclear and somatic reprogramming research. The phenomenon of nuclear reprogramming was observed in the early 1960s via SCNT and heterokaryon experiments. In 1997, the cloning of Dolly served as a monument of mammalian cloning and addressed the plasticity characteristic in mammals’ genomes. Furthermore, the cloning of the Macaque monkey counted as a recent breakthrough in the nuclear reprogramming field. Along with the advances of stem cell biology, the unraveling of pluripotent network formation and new culture approaches (e.g., 2i medium and feeder-free) accelerated iPSC development and further allowed for capturing and maintaining EPS cells (expanded potential stem cells) in vitro. Another crucial breakthrough was the production of the hiPSC-derived RA-responsive FGC formation. All of these advances contributed to the first clinical trials of hiPSC-derived retinal cell transplants.

**Figure 2 cells-10-02888-f002:**
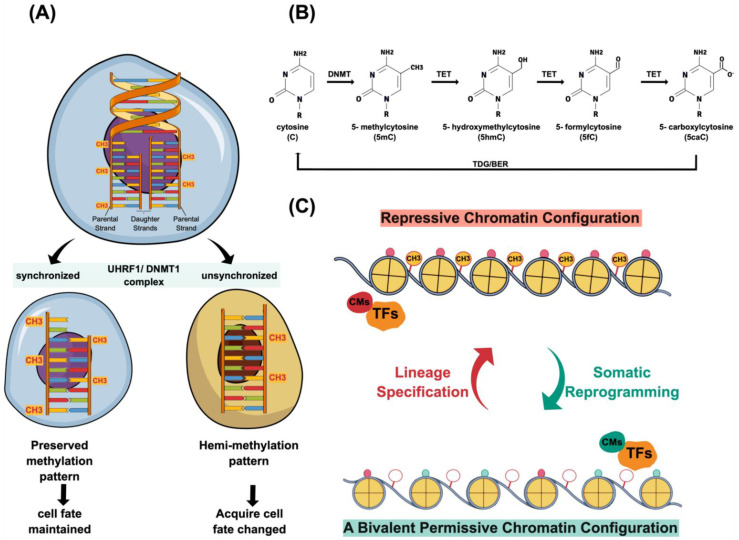
Diverse mechanisms coordinate genome plasticity. (**A**) The replication-mediated passive DNA demethylation process results in cell fate change. The newly replicated strands do not share the same methylation pattern as seen in their parental DNA molecule. To faithfully maintain the methylation pattern, the complex of UHRF1/DNMT1 is recruited to perform the de novo methyl-transfer activity on the newly synthesized strands. This uncoupled methyl-transfer reaction from DNA replication offers an opportunity to diversify the cell fate. In the scenario of an unsynchronized event, the unmethylated daughter strand creating a hemimethylated pattern may allow its underlying regulatory elements to be susceptible to the signal inputs from the surrounding environment. Therefore, cell division event creates two daughter cells bearing different cell states, which is feasible for diversifying cell fates. (**B**) The Tet-mediated active DNA demethylation mechanism. Tet1, Tet2, and Tet3 belong to the Tet family of DNA demethylase. These Tets catalyze 5-hydroxymethylcytosine (5mC) through a series of oxidation processes, and generate a set of intermediated products, including 5-hydroxymethylcytosine (5hmC), 5-formylcytosine (5fC), and 5-carboxylcytosine (5caC). Furthermore, the thymine DNA glycosylase (TDG) mediates the excision of 5fc and 5caC. Eventually, the base excision repair (BER) mechanism restores the cytosine nucleotide in the demethylation process. (**C**) Chromatin-mediated cell fate change. In addition to the DNA methylation pattern impacting the epigenetic configuration, chromatin modifications also play a pivotal role in regulating the epigenetic landscape. A two-way relationship between transcription factor binding and chromatin structure modification further shapes its epigenetic landscape. Hence, the transcriptional network, chromatin structure, and DNA methylation pattern work together to establish a unique state of epigenomic configuration representing an individual cell fate. TFs—transcription factors; CMs—chromatin modifiers.

**Figure 3 cells-10-02888-f003:**
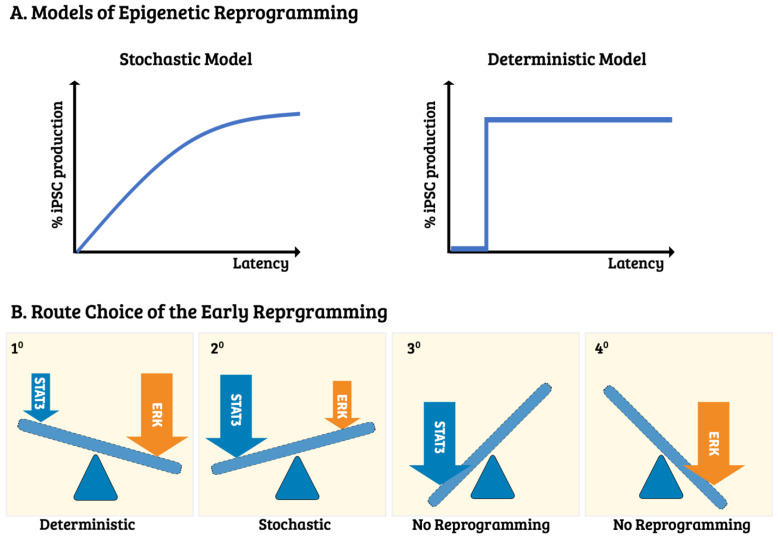
Models of epigenetic reprogramming and their potential mechanism in early route choice. (**A**) Somatic reprogramming takes its action mode at the beginning of epigenetic remodeling. Two well-recognized patterns of reprogramming kinetics, namely stochastic and deterministic models, are used to gain pluripotency-competent cells during somatic reprogramming. Latency in the X-axis indicates the required time or the number of cell divisions to acquire pluripotency, whereas the Y-axis shows the percentage of pluripotent cell numbers. The left panel displays the stochastic model. All of the candidate cells after certain cell divisions have an equal chance of undergoing reprogramming. Therefore, the acquisition of pluripotency is a random event [2,64]. The right pane shows the deterministic model where only the specific cells with a privileged state can overcome the epigenetic barriers set up by the deterministic process. Therefore, it is synchronized with a fixed latency to surpass any encountered hurdle [65,66,67]. (**B**) Liu et al. argued that the onset route choice of reprogramming depends on the balance of the EGFR/ErbB2 and LIFR/gp130 pathways. In the 1° panel, input signals more on the EGFR/ErbB2 pathway endow a group of synchronized cells to surpass the reprogramming barrier. In contrast, if they rely more on the LIFR/gp130 pathway, the stochastically unsynchronized cells gain pluripotency at their own pace (2° panel). Eliminating the inputs from either of the signal pathway fails to achieve pluripotency (3° and 4° panels). Thus, different modes of operation can reversibly act on the same MEF cells through modulating the activity of STAT3 and Erk.

**Figure 4 cells-10-02888-f004:**
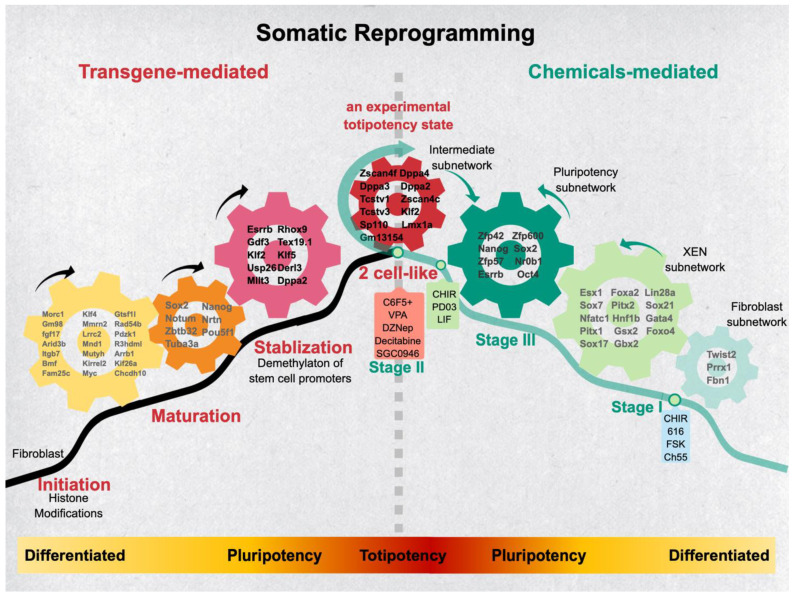
Distinct regulatory networks control epigenetic transits in tgiPSC and CiPSC. Genes and their correlated networks engage cell fate transit during reprogramming from MEF to iPSC. The **left** panel represents the roadmap of transgene-mediated iPSC (tgiPSC) formation, including the initiation, maturation, and stabilization phases. In this case, the first encountered obstacle is unlocking the fibroblast fate for MET. By contrast, chemical-based reprogramming adopts a different route (CiPSC; **right** panel). Instead of proceeding to MET, it first adopts the XEN-like cell fate by expressing the XEN-subnetwork genes. Subsequently, the network relays to the 2C-like program before acquiring the pluripotency circuitry. Although several recent reports have observed transient 2C-like programs during the process of tgiPSC, more evidence is needed in order to support whether such a 2C-like transit state is mandatory for tgiPSC formation [113,114,115]. Nevertheless, the common theme for both approaches shows that stepwise cell fate transitions are achieved through relaying to different regulatory networks during its progression. Essential markers and specific hallmarks of each phase of reprogramming are indicated.

## Data Availability

Not applicable.

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
