# Peer review of "Somatic Reprogramming—Above and Beyond Pluripotency"

_cells, 2021, doi:10.3390/cells10112888_

Round 1
Reviewer 1 Report
The Authors summarize and discuss the aspects of somatic reprogramming ranging from the molecular principles and mechanisms to the first clinical trials on iPSCs. The review is well organized and comprehensive.
The only issue I have is that the manuscript requires an extensive English editing for style and usage (there are several incorrect constructions and expressions that are not used in English).
Author Response
The Authors summarize and discuss the aspects of somatic reprogramming ranging from the molecular principles and mechanisms to the first clinical trials on iPSCs. The review is well organized and comprehensive.
The only issue I have is that the manuscript requires an extensive English editing for style and usage (there are several incorrect constructions and expressions that are not used in English).
Response:
Thank you very much for the comments. I will seek English-editing assistance before proceeding to publication.
Reviewer 2 Report
Meir et al. provided a review about reprogramming.
In this report the authors highlight the importance of the epigenetic changes occurring during reprogramming. They describe the steps involved in the re-acquisition of pluripotency, comparing also the different features of tgiPSC and CiPSC. Finally, the authors conclude reporting some interesting experiments aimed at overcoming the pluripotency, by generating totipotent-like cells, and discuss the potential use of pluripotent stem cells for clinic purposes including advantages and disadvantages of this cellular model.
The topic is very interesting and appropriate. A review discussing the link between reprogramming and epigenetics could be very useful for the readers, however critical changes are required before proceeding to publication.
1. Since the transgene-based strategy includes many different approaches, such as integrative (lentiviral, retroviral) and non-integrative methods (integrative-defective viruses, episomal and RNA delivery etc..), I would suggest to the authors to include a brief description about these approaches in the first part of section 4, clarifying pros and cons of each approach over the others.
2. In the timeline reported in Figure 1 I would add the generation of the haplobanks, because they could help to reduce the time and the costs required for the generation of an autologous hiPSC cell line, the risk of immune reactions and the use of the immunosuppressive treatment after a heterologous hiPSC hiPSC-derived cells transplantation, with great benefits for patients.
3. In Figure 2 please modify the representation of the hemi-methylated pattern, adding a picture of hemi-methylated DNA in the cell nucleus.
4. Please define the acronym “EPSCM”, I suppose that the authors refer to the culture medium for the EPSCs, but they need to specify it.
5. The last paragraph in section 7 has a different style, please modify it according to the publisher’s guidelines.
6. The manuscripts should be extensively proof-read by a native English speaker, before publication and the manuscript edited to streamline the flow and ease the reading of the text. In several instances, indeed, I found it hard to understand what the authors actually meant.
Author Response
Meir et al. provided a review about reprogramming.
In this report the authors highlight the importance of the epigenetic changes occurring during reprogramming. They describe the steps involved in the re-acquisition of pluripotency, comparing also the different features of tgiPSC and CiPSC. Finally, the authors conclude reporting some interesting experiments aimed at overcoming the pluripotency, by generating totipotent-like cells, and discuss the potential use of pluripotent stem cells for clinic purposes including advantages and disadvantages of this cellular model.
The topic is very interesting and appropriate. A review discussing the link between reprogramming and epigenetics could be very useful for the readers, however critical changes are required before proceeding to publication.
- Since the transgene-based strategy includes many different approaches, such as integrative (lentiviral, retroviral) and non-integrative methods (integrative-defective viruses, episomal and RNA delivery etc..), I would suggest to the authors to include a brief description about these approaches in the first part of section 4, clarifying pros and cons of each approach over the others.
Response:
Yes. I included the pros and cons of different transgene systems for iPSC production in the first part of section 4 (in red).
- In the timeline reported in Figure 1 I would add the generation of the haplobanks, because they could help to reduce the time and the costs required for the generation of an autologous hiPSC cell line, the risk of immune reactions and the use of the immunosuppressive treatment after a heterologous hiPSC hiPSC-derived cells transplantation, with great benefits for patients.
Response:
Yes. I added three references regarding the generation of the haplobanks in Figure 1.
- In Figure 2 please modify the representation of the hemi-methylated pattern, adding a picture of hemi-methylated DNA in the cell nucleus.
Response:
Yes. I revised Figure 2 with the hemi-methylated DNA in the cell nucleus.
- Please define the acronym “EPSCM”, I suppose that the authors refer to the culture medium for the EPSCs, but they need to specify it.
Response:
Yes. I addressed the abbreviation of EPSCM in the text (page 18) and keywords (page 1). It refers to the medium containing the cocktail of inhibitors and LIF as Expanded Potential Stem Cell Medium.
- The last paragraph in section 7 has a different style, please modify it according to the publisher’s guidelines.
Response:
Yes, I revised the font in the last paragraph of section 7 according to publisher’s guidelines.
- The manuscripts should be extensively proof-read by a native English speaker, before publication and the manuscript edited to streamline the flow and ease the reading of the text. In several instances, indeed, I found it hard to understand what the authors actually meant.
Response:
Thank you very much for the comments. I will seek English-editing assistance before proceeding to publication.
Reviewer 3 Report
This is a good review and I read it with interest. Its title is misleading in part, as reprogrammed cells that can be "localized" beyond pluripotency are discussed in the two final paragraphs of the review, and honestly the derivation of the germ cells has little to do with the concept of reprogramming (rather, that it is part of the "programming"), but the part on epigenetics and its deep involvement in the reprogramming is valuable, and offers the bases to understand why covering the small step between pluripotent and totipotent is so hard.
The corpus of references is huge but not particularly up to date. Some keystone references are also missing (reprogramming with 2F, 1F, other TFs, like Nanog, Lin28; Jaenish, Schöler, Hochdenlinger and others), and the authors did not stress enough that different reprogramming techniques yields different efficiency of reprogramming and different levels of epigenome erasure (for example, see Churko et al., Nat Biomed Eng, 2017; Roost et al., Nat Comm, 2017; Bar & Benvenisty, EMBO J, 2019). Again on the literature front, the authors should mention other SCNT approaches, besides Gurdon's (e.g. Wilmut et al, 1997; Wakayama et al., 1998; Zhou et al., 2003; ...), for completeness.
Going to specific points, some sentences should be revised/rephrased, as they are not completely clear or correct, a list is given below.
pag1, abstract, the third last sentence ( "Thus, along with ...") is not clear.
pag1, introduction, third last sentence ( "One of the main ways to reconfiguration ...") is grammatically not correct.
pag3, line1: EPS abbreviation only explained later, pag14.
pag5, line 27: PRC2 the polycomb complex name should be at least given before the abbreviation
pag6, the first block (from "the pluripotency factors...") should be revised: as it is wriotten, it seems to me that 1. the pluripotency factors are only Oct4, Sox2, Tbx3 and Nanog (there are many others, like Utf1, Essrb, Rex1, Zic, dppa, lncRNAs, ... one may say that the traditional CORE pluripotency network is made by Oct4, Nanog and Sox2, but the real network is much larger); 2. that they are only germ layer specifiers (not only!); 3. that pluripotency is maintained because these 4 TFs antagonize each others (indeed, they do not only antagonize each others, they also cross-activate each others, and at the same time they antagonize many other developmental genes, to prevent differentiation).
pag6/7, the authors mentioned that the the mechanisms behind the reacquisition of pluripotency induced by SCNT and nuclear fusion are "not as evident" as those from iPSCs. Thisis not completely correct. The papers quoted about these techniques are very old. many more had come that also explain how reprogramming was possible using SCNT and nuclear fusion from the mechanistic point of view.
Pag8, legend fig3: maybe the first sentence should be rephrased (I think that "to choose" requires rational thinking).
pag10, lower third of the page: the evidences discussed by Chung and colleagues (Chung et al., PLOS ONE, 2014) about the stochastic phase of reprogramming gained from a single cell approach should also be discussed here.
pag15, second sentence of paragraph 6: I think that SCNT is reminiscent of fertilization, and not vice-versa...!
A couple of minor considerations: the use of the first person "I" should be avoided (use "we" instead); the abbreviation list is very useful, but far from being complete; the last block of the sixth paragraph turned out in a different font.
Author Response
This is a good review and I read it with interest. Its title is misleading in part, as reprogrammed cells that can be "localized" beyond pluripotency are discussed in the two final paragraphs of the review, and honestly the derivation of the germ cells has little to do with the concept of reprogramming (rather, that it is part of the "programming"), but the part on epigenetics and its deep involvement in the reprogramming is valuable, and offers the bases to understand why covering the small step between pluripotent and totipotent is so hard.
- The corpus of references is huge but not particularly up to date. Some keystone references are also missing (reprogramming with 2F, 1F, other TFs, like Nanog, Lin28; Jaenish, Schöler, Hochdenlinger and others), and the authors did not stress enough that different reprogramming techniques yields different efficiency of reprogramming and different levels of epigenome erasure (for example, see Churko et al., Nat Biomed Eng, 2017; Roost et al., Nat Comm, 2017; Bar & Benvenisty, EMBO J, 2019).
Response:
Yes. I included this part of the discussion in the first part of section 4 (page 12-13; in red).
- Again on the literature front, the authors should mention other SCNT approaches, besides Gurdon's (e.g. Wilmut et al, 1997; Wakayama et al., 1998; Zhou et al., 2003; ...), for completeness.
Response:
Yes. I included these pioneers' works in the reference list.
Going to specific points, some sentences should be revised/rephrased, as they are not completely clear or correct, a list is given below.
3. pag1, abstract, the third last sentence ( "Thus, along with ...") is not clear.
Response:
I meant to say, “With the prior fruitful studies in SCNT and cell fusion experiments, iPSC finds its place and implicates that the differentiated somatic epigenome remains plasticity for re-gaining the pluripotency and further stretchable to reach a totipotency-like state.” Regarding this sentence, I will further discuss it with my English-editing service to make it clear. Thanks for pointing it out.
- pag1, introduction, third last sentence ("One of the main ways to reconfiguration ...") is grammatically not correct.
Response:
Corrected as follows:
“One of the main ways to reconfigure the epigenetic landscape is through DNA replication during cell division.” Thanks for pointing it out.
- pag3, line1: EPS abbreviation only explained later, pag14.
Response:
I put the EPS (Expanded Potential Stem Cells) abbreviation in the keywords and Figure 1’s legend, which is the first sentence encountered to the EPS jargon (in red).
6. pag5, line 27: PRC2 the polycomb complex name should be at least given before the abbreviation
Response:
Yes. I revised and placed “the polycomb complex (PRC2)” in page 7 (in red).
7. pag6, the first block (from "the pluripotency factors...") should be revised: as it is wriotten, it seems to me that 1. the pluripotency factors are only Oct4, Sox2, Tbx3 and Nanog (there are many others, like Utf1, Essrb, Rex1, Zic, dppa, lncRNAs, ... one may say that the traditional CORE pluripotency network is made by Oct4, Nanog and Sox2, but the real network is much larger); 2. that they are only germ layer specifiers (not only!); 3. that pluripotency is maintained because these 4 TFs antagonize each others (indeed, they do not only antagonize each others, they also cross-activate each others, and at the same time they antagonize many other developmental genes, to prevent differentiation).
Response:
I rearranged the sentences as follows:
“The traditional core pluripotency factors, including Oct4, Sox2, Tbx3, and Nanog, were recognized as the members of germ layer specifiers. They function in a manner of precarious balance through antagonizing and cross-activating each other to reach a pluripotent state. The antagonistic effect halts the pluripotent state from falling into any germ layer lineages (Loh et al., 2011). This idea had been further fortified in cell fate change by overexpression or knockdown of those lineage specifiers.”
- pag6/7, the authors mentioned that the the mechanisms behind the reacquisition of pluripotency induced by SCNT and nuclear fusion are "not as evident" as those from iPSCs. This is not completely correct. The papers quoted about these techniques are very old. many more had come that also explain how reprogramming was possible using SCNT and nuclear fusion from the mechanistic point of view.
Response:
I revised the sentence as follows:
“Although SCNT and heterokaryon experiments were performed much earlier than iPSC, their pluripotency re-acquisition process and the corresponding mechanisms were revealed from recent studies (Matoba et al., 2018). Here, we only focus on the current progress in unraveling the transgene-based reprogramming mechanism, even though it is very intriguing that both SCNT and heterokaryon changing in chromatin assembling is independent of DNA replication.”
- Pag8, legend fig3: maybe the first sentence should be rephrased (I think that "to choose" requires rational thinking).
Response:
I revised the sentence as follows:
“Somatic reprogramming takes its action mode at the beginning of epigenetic remodeling.”
- pag10, lower third of the page: the evidences discussed by Chung and colleagues (Chung et al., PLOS ONE, 2014) about the stochastic phase of reprogramming gained from a single cell approach should also be discussed here.
Response:
Yes. I included a paragraph (in red) to discuss Chung and colleagues’ work (the last paragraph on page 11).
- pag15, second sentence of paragraph 6: I think that SCNT is reminiscent of fertilization, and not vice-versa...!
Response:
I revised the sentence as follows:
“After fertilization, the sperm pronucleus is triggered to reprogram by maternal factors in the ooplasm. Similarly, both SCNT and heterokaryon exposed their differentiated nuclei in the ooplasm or ESC’s cytosol, where the plasma factors initiated the process of reprogramming.”
- A couple of minor considerations: the use of the first person "I" should be avoided (use "we" instead); the abbreviation list is very useful, but far from being complete; the last block of the sixth paragraph turned out in a different font.
Response:
Yes, I revised the first person “I” to “we”. The font is also corrected.